# Frequency of CYP3A5 Genetic Polymorphisms and Tacrolimus Pharmacokinetics in Pediatric Liver Transplantation

**DOI:** 10.3390/pharmaceutics12090898

**Published:** 2020-09-22

**Authors:** Jefferson Antonio Buendía, Esteban Halac, Andrea Bosaleh, María T. Garcia de Davila, Oscar Imvertasa, Guillermo Bramuglia

**Affiliations:** 1Department of Pharmacology and Toxicology, Faculty of Medicine, University of Antioquia, Medellin 050010, Colombia; 2Liver Transplant Service, J.P. Garrahan Hospital, Buenos Aires C1245AAM, Argentina; ehalac@gmail.com (E.H.); imventarzaoscar@gmail.com (O.I.); 3Pathology Service, J.P. Garrahan Hospital, Buenos Aires C1245AAM, Argentina; apbosaleh@yahoo.com.ar (A.B.); gdedavila@gmail.com (M.T.G.d.D.); 4Faculty of Pharmacy and Biochemistry, University of Buenos Aires, Buenos Aires C1113, Argentina; gfbramuglia@gmail.com

**Keywords:** tacrolimus, CYP3A5, liver transplant, pharmacokinetics

## Abstract

The evidence available in the pediatric population is limited for making clinical decisions regarding the optimization of tacrolimus (TAC) in pharmacotherapy. The objective of this study was to estimate the frequency of CYP3A5 genetic polymorphisms and their relationship with tacrolimus requirements in the pediatric population. This was a longitudinal cohort study with a two-year follow-up of 77 patients under 18 years old who underwent a liver transplant during the period 2009–2012 at the J.P. Garrahan Pediatric Hospital. Tacrolimus levels from day five up to two years after the transplant were obtained from hospital records of routine therapeutic drug monitoring. The genotyping of CYP3A5 (CYP3A5*1/*3 or *3/*3) was performed in liver biopsies from both the donor and the recipient. The frequency of CYP3A5*1 expression for recipients was 37.1% and 32.2% for donors. Patients who received an expresser organ showed lower Co/dose, especially following 90 days after the surgery. The role of each polymorphism is different according to the number of days after the transplant, and it must be taken into account to optimize the benefits of TAC therapy during the post-transplant induction and maintenance phases.

## 1. Introduction

Tacrolimus (TAC) is a calcineurin inhibitor widely used in solid organ transplantation. TAC has a narrow therapeutic margin and a large intra- and inter-individual variability [1,2]. Incidence of rejection and adverse effects remain as problems despite therapeutic drug monitoring of TAC [3]. There is growing interest in developing markers that will allow for an individual treatment with TAC. Within this group of potential biomarkers, we find the single nucleotide polymorphisms of CYP3A5 [3,4,5]. This enzyme has a highly polymorphic expression with at least 11 single nucleotide polymorphisms (SNPs) documented [3]. The most studied SNP is the transition from adenine to guanine at the position 6986-intron 3-CYP3A5 gene (rs776746), also known as CYP3A5*1. This allele is associated with high levels of CYP3A5-mRNA and fully functional CYP3A5-protein [6,7]. The Caucasian population expresses CYP3A5*1 between 10–40%, while the Asian population expresses it between 50–70% [8]. CYP3A5*1 (homozygotes and heterozygotes) expressers require much higher daily doses of TAC as well as more time to reach its desired serum levels. Furthermore, expressers have three times the risk of acute rejection within the first month after transplant than non-expressers [9].

After a liver transplant, the simultaneous expression of CYP3A5*1 in both the intestine and the implanted liver may occur [3]. In previous studies in the adult population, we showed that this interaction does occur. The expression of CYP3A5*1 present in the liver donor has a great impact on TAC levels adjusted by dose in long-term concentrations, while the expression of this SNP in the receiver also has a great impact, but only after transplantation [8]. However, its kinetics and pharmacodynamics are very different when comparing pediatric and adult populations. This can be explained by the greater variability of specific enzymes, which are acquired by the child during growth and alter the clinical response to TAC [3]. The evidence available in the pediatric population is limited for making clinical decisions regarding the therapeutic optimization of TAC. Thus, it is essential to generate more information to optimize and customize monitoring strategies for liver transplants in this population. The objective of this study was to estimate the frequency of CYP3A5 genetic polymorphisms and their relationship with pharmacokinetics in pediatric liver transplantation.

## 2. Materials and Methods

A longitudinal study was conducted in 77 patients under 18 years old after liver transplantation during the period 2009–2012 at the J.P. Garrahan Pediatric Hospital (JPGPH).

Patients with full or partial liver grafts from either living or cadaveric donors were included. All patients were receiving tacrolimus with or without steroids and with or without mofetil mycophenolate (MMF). We excluded HIV infected patients who suffered an early death before receiving an immunosuppressive regimen with TAC immediately after surgery and patients with partial or total loss of medical records.

### 2.1. Dosage and Treatment Scheme

Patient information was collected immediately after the liver transplantation. The immunosuppression scheme used on the subjects according to the Clinical Practice Guidelines of JPGPH for patients after liver transplantation is described below. During the induction phase, all patients received basiliximab. Patients who weighed less than 30 kg received a 10 mg/dose, and those who weighed more than 30 kg received a 20 mg/dose. Both doses were administered as an intravenous bolus, the first one within 8 h after reperfusion of the graft and the second one on the fourth day after surgery. TAC was dispensed in the maintenance phase, which started 24 h after reperfusion. The initial oral regimen was 0.1 mg/kg/day every 12 h. After, the dose of TAC was adjusted to tacrolimus blood levels, liver parameters, kidney function, and the viral load of Epstein Barr Virus (EBV) [10]. In patients without infectious activity (viral load less than 4000 copies/μg DNA) and a creatinine clearance less than the expected range for their age, the initial desired TAC blood levels were 8–12 ng/mL during the first month after transplantation and then 5, 6, 7, and 8 ng/mL until a year after transplantation had passed [10]. We proceeded with a quick immunosuppression reduction in those patients with viral loads above 4000 copies/μg DNA in two consecutive samples or clinical evidence of EBV infection. No antiviral therapy was implemented. In patients who developed renal toxicity, regardless of viral load, monitoring of TAC was decreased to 25%. In those cases, MMF was added as rescue therapy with an initial dose of 20 mg/kg/day, and then it was increased up to 40 mg/kg/day after a week of treatment.

### 2.2. Monitoring and Quantification of Tacrolimus Blood Levels

TAC levels from day five up to the second year after the transplant were obtained from hospital records of routine therapeutic drug monitoring. The values recorded correlated to monitoring blood levels from samples drawn prior to the morning dose or Co (concentration measured in *t* = 0 before the first dose of the drug).

The quantification of TAC was performed by chemiluminescence immunoassay by Abbott’s Architect i1000 according to the manufacturer’s instructions. The low quantification limit was 2.0 ng/mL, and linearity was observed between 2–30 ng/mL. The variation coefficient for quality control samples was lower than 6%.

### 2.3. Collected Information

Demographic information (date of birth, gender), anthropometric data (weight, height), indication of transplant, post-transplantation follow-up time, current medication and doses, concomitant medications, amount of transplanted graft, amount of postsurgical days, and data related to the donor type were collected. We registered clinical laboratory results including hematology (hemoglobin, hematocrit, red blood cells, white cells and platelets, prothrombin time (PT), activated partial thromboplastin time (aPTT), and thrombin time (TT) and clinical chemistry results (creatinine, urea nitrogen, total and direct bilirubin, alkaline phosphatase, alanine aminotransferase (GPT or ALT), aspartate aminotransferase (GOT or AST), gamma glutamyl transpeptidase (GGT), and albumin.

### 2.4. DNA Isolation and Genotyping

The genotyping of CYP3A5 was performed in liver biopsies of both the donor and the recipient. The donor’s DNA was obtained from liver biopsies or surgical specimens obtained from the pathology service at the JPGPH. Each of them was tissue-fixed in formalin-buffer, embedded in paraffin, and sectioned 10 microns thick.

DNA extraction was performed using commercial kits QIAamp DNA Blood Kit and QIAamp DNA FFPE Tissue following the manufacturer’s instructions. We obtained from 20 to 100 ng DNA in each case. CYP3A5*3 (rs776746) polymorphism was detected by PCR and directly sequenced. Patients with variants CYP3A5*1/*1 or CYP3A5*1/*3 were called “expressers”, while those with variants CYP3A5*3/*3 were called “non-expressers”.

### 2.5. Ethical Aspects

A proper informed consent was signed by a parent or legal guardian before starting any specific evaluations. The study was approved by the office of Teaching and Research at the JPGPH (Code 740 21/08/12) and by the Ethics Committee of the Faculty of Pharmacy and Biochemistry, at the University of Buenos Aires (Code 930 21/03/14).

### 2.6. Statistical Analysis

We compared daily doses of TAC, Co (TAC levels before the morning dose), and Co/dose (concentration adjusted by dose) according to CYP3A5*1 allele expression between donors and recipients. All values were expressed as mean ± standard deviation. The U Mann-Whitney test was used to determine differences among continuous variables in the groups. The chi-square test was used to analyze differences among discrete variables. All analyses were performed using STATA 11.0^©^ (Lakeway Drive, College Station, TX, USA).

## 3. Results

We evaluated 77 pediatric patients medicated with TAC during the first two years after transplantation. Table 1 shows the characteristics of the population studied. We observed 45 patients (58.44%) with adverse events associated with tacrolimus, 51 patients (66.23%) had at least one acute cellular rejection episode, and eight patients died (10.39%) during follow-up.

CYP3A5*1 expression was 37.1% in recipients and 32.2% in donors. There were no statistically significant deviations in the distribution of polymorphisms according to the Hardy-Weinberg principle (*p* > 0.05).

A total of 3670 blood concentrations of TAC were analyzed during the study period with a mean of 47.8 samples per patient. We observed a greater difference in expresser recipients compared with non-expressers, especially in the first two weeks after surgery, and those differences tended to reduce over time (Figure 1).

When we adjusted the dose by concentrations according to the genotype of the donor, those who received an expresser organ showed a lower Co/dose, especially 90 days after surgery (Figure 2). A statistically significant reduction of 0.00063 ng/mL mg/kg/day in the Co/dose was observed compared with those receiving a non-expresser organ (*p* = 0.001).

## 4. Discussion

CYP3A5 polymorphisms have a differential impact in the pharmacokinetics of tacrolimus according to its expression in donors and recipients. In contrast to previous studies, this is the first study in pediatric patients that evaluates the effect of polymorphisms on TAC pharmacokinetics in the long term. Other studies considered shorter periods and generally did not include the Hispanic population.

Patients with CYP3A5 allele A (CYP3A5*1 or wild type) have a normal splicing of all 13 exons in this gene. This results in a normal transcript and a production of high levels of mRNA, which in turn expresses the enzyme metabolizing TAC. Patients with allele G (CYP3A5*3) have a point mutation (A/G) resulting in the insertion of an inappropriate 3B “exon” within the transcript. This new exon introduces an early termination codon, leading to a non-functional protein fragment [11]. The frequency of expressers (CYP3A5*1) in our study was reported to be intermediate within the ranges of Asian (33% to 66%) and Caucasian (9% to 15%) populations. These estimates were consistent with previous results in studies in Argentinean renal transplant patients, which reported values ranging from 9% to 27% [12,13,14]. These differences between Caucasian and Asian frequencies reveal the genetic diversity present in Latin America as a result of the colonial era, as well as the African (slaves in the 19th century) and post-independence immigrants (the majority coming from Spain, Italy, France, and eastern Europe) [15].

Similar results have been found in studies focused on the frequency of variations in other genes related to anti-neoplastic metabolism [16]. Continuing to build this pharmacogenetic map in Latin America will improve the understanding of the variations in the metabolism and the effect of various drugs without the need to extrapolate results obtained from other populations.

In liver transplant patients, both donors and recipients carrying the CYP3A5 polymorphisms are associated with changes in the pharmacokinetics of TAC. Nevertheless, the role of each polymorphism is different according to the number of days after transplantation. We have shown that the recipient CYP3A5 genotype plays a more important role than the donor genotype. Recipients with CYP3A5*1 achieved lower blood concentrations of TAC and lower dose-adjusted concentrations despite the medical pharmacotherapeutic follow-up (based on adjusting the blood concentrations to the reference therapeutic margins). These findings were consistent with a recent study of 64 post-transplant children with a one year follow-up [17]. It was shown that lower dose-adjusted (*p* < 0.05) concentrations were required in patients who were expressers, without any correlation with donor genotype, especially in the first seven days after transplantation [17].

To recognize the role played by the recipient CYP3A5 genotype in the first weeks after transplantation, it is essential to avoid excessive dose increases in patients who are expressers of this genotype [18].

On the other hand, the donor genotype alters the kinetics of TAC, significantly increasing with time after transplantation. The effect of CYP3A5 expression on the recipient is an increased hepatic clearance of the liver implanted with the polymorphism. This tendency was evidenced in our study, where a reduction of dose-adjusted concentrations was documented as statistically significant 60 to 90 days after surgery. This observation might be related to the time needed by the organ to recover from the ischemia and reperfusion injuries, regeneration, and graft growth as months after transplantation pass [19]. Our results indicate the importance of knowing the genotype present in the organ before being transplanted. During the ambulatory follow-up, it is a priority to select patients with greater hepatic clearance who will get a lower concentration and who may require different medical follow-up to avoid sub-immunosuppression.

Our study has limitations primarily due to its retrospective nature. Among them, some are due to the misclassification of patients either by memory bias or problems with recording information in clinical histories (by omission or incorrect recording). These biases could be minimized by always obtaining information from primary registers (physical or electronic medical history) and double-checking with other clinical records (nursing records and hospital pharmacy records). Additionally, we only analyzed concentrations per patient at time 0 (C0) given the fact that we used hospital therapeutic monitoring data, and for TAC, this concentration is used for clinical monitoring of these drugs. Also, the effect of other variables on the pharmacokinetics of TAC, such as age, drug interaction, and length of the event related to dose-adjusted concentrations, were not evaluated and should be analyzed in further studies.

In conclusion, patients after liver transplantation—both donors and recipients—carrying CYP3A5 polymorphisms are susceptible to suffering changes in TAC pharmacokinetics. However, the role of each polymorphism is different according to the number of days after transplantation, and it must be taken into account to optimize the benefits of TAC therapy during the post-transplant induction and maintenance phases.

## Figures and Tables

**Figure 1 pharmaceutics-12-00898-f001:**
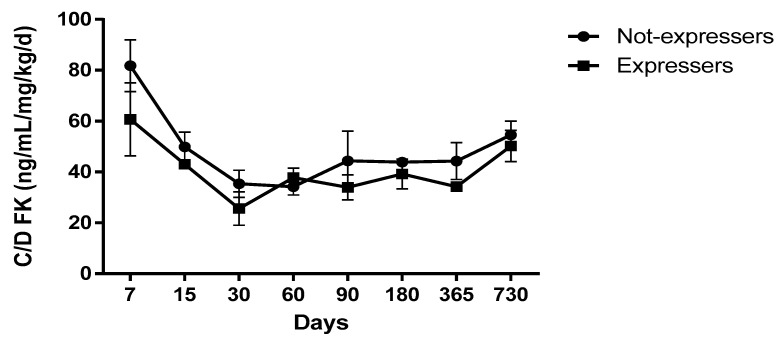
Co/dose of tacrolimus (TAC) according to the recipient genotype.

**Figure 2 pharmaceutics-12-00898-f002:**
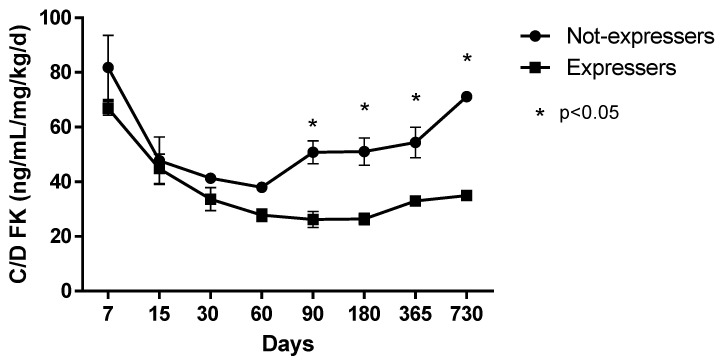
Co/dose of TAC according to the donor genotype.

**Table 1 pharmaceutics-12-00898-t001:** Characteristics of the studied population (*n* = 77).

Feature	*n* (%)
Female	46 (59.74)
Age at transplantation (years, ±SD)	5.32 (5.42)
Weight (kg, ±DE)	21.84(17.89)
Origin	
Argentina	64(83.11)
Bolivia	2(2.60)
Paraguay	9(11.69)
Other	2(2.59)
**Primary disease**	
Biliary atresia	32(41.55)
Fulminant hepatitis	16(20.77)
Autoimmune hepatitis	11(14.28)
Hepatoblastoma	8(10.38)
Others	10(12.98)
**Kind of Donor**	
Cadaveric	55(71.42)
Live	22(28.57)
**Kind of Graft**	
Full	26(33.76)
Technical variant	51(66.23)

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
