# Peer review of "Frequency of CYP3A5 Genetic Polymorphisms and Tacrolimus Pharmacokinetics in Pediatric Liver Transplantation"

_pharmaceutics, 2020, doi:10.3390/pharmaceutics12090898_

Round 1

Reviewer 1 Report

I think the paper requires extensive improvements.

For example, I am sorry but could not undersand what they mean by "Co/dose"

Author Response

For example, I am sorry but could not undersand what they mean by "Co/dose"

Response: Correction accepted.  An operative definition of Co and Co / dose is placed

Line 119-120 " Statistical analysis : We compared daily doses of TAC, Co (TAC levels prior to the morning dose) and Co/dose (concentration adjusted by dose) .....

Reviewer 2 Report

In this article, Buendia and coworkers report the frequency of CYP3A5 genetic polymorphisms and Tacrolimus treatment in Pediatric Liver Transplantation. 

There are several issue with this work that requires major revision:

Lines 2-3: The title needs revision.

Lines 4-5: A * is missing to show the corresponding author.

Lines 6-13: The affiliations need revision. Emails of the authors are missing. After each affiliation there is no need of commas or dots.

Lines 11-13: The correspondence information is not written in the right way.

Lines 15-27: Abstract does not follow the instructions of the journal. This must be written without headings. 

Line 28: Keywords (3-10) should be included between the Abstract and the Introduction.

Line 37: Check language-grammar. There should not be comma at the rs Number of a SNP.  

Line 38: Check grammar. The allele is associated not the SNP. 

Line 41: Non-expressers. Also, non-expressers are homozygous for *3 (CYP3A5*3/*3). Check phrasing.

Lines 44-45: Check phrasing.

Line 55:  Check Punctuation.

Line 56: Check phrasing. 

Line 57: Check Punctuation.

Line 58: Section 2- Materials and Methods not Methods and Materials.

Line 59: Check phrasing.

Line 60: Check phrasing.

Line 61 + 62: Check phrasing.

Line 64: Check phrasing.

Line 65: Check phrasing and Grammar. 

Line 64-65: Change syntax.

Line 67: Check Grammar. 

Line 69: patients should not be in capital.

Line 74: tacrolimus blood levels.

Line 76: ug DNA not DNA ug. Delete initial. Tacrolimus blood levels.

Line 77: Delete ΄΄:΄΄. Check Punctuation. Check phrasing.

Line 79: Numbers 1 to 9 are written in full, except if part of a measurement (6–8 mL) or in the experimental/materials/methods section.

Line 80: No meaning.

Line 82: Check Grammar. 

Line 95: Check Grammar. 

Line 97: Check phrasing (amount). 

Line 98: Check phrasing (amount). 

Line 99: haematological. Red blood cells.

Line 100: White blood cells. RINàDefine abbreviations. 

Line 104: Check Grammar.

Lines 11-112: Check Grammar, phrasing of the whole sentence.

Line 123: Check Grammar. 

Line 127: Check grammar.

Line 129:  Check phrasing.

Line 130: Change syntax

Line 131: Check phrasing.

Line 134: Check Punctuation. Change receptor expressers to expressers recipients.

Line 135: Check phrasing, Grammar and syntax.

Line 139: Delete ΄΄-΄΄.

Line 141: Check Punctuation. A full stop should be placed at the end of the title.

Table 1: Check ΄΄origen΄΄, change phrasing (full, technical variant).

Figure 1: Check grammar. Non-expressers.

Line 145: Check grammar.

Fugure 2: Check grammar. Non-expressers.

Line 150: Check grammar, phrasing.

Line 153: Check grammar.

Line 154: CYP3A5 is a gene, not a variant (do not use rs number).

Line 155: CYP3A5 is a gene, not a variant (do not use rs number), write genotype.

Line 158: Delete CYP3A5.

Line 163: references 15, 16, 17 should be placed at the end of the sentence. 

Line 165: Check grammar. Reference 18 should be placed at the end of the sentence. 

Line 166: East (with capital E).

Line 167: Check grammar. (pharmacogenetic studies not pharmacogenetics).

Line 169: CYP1A12C, CYP1A21F,CYP3A41B, CYP2D62 are not genes but variants.

Lines 170-173: The sentence does not make any sense.

Line 173: Check grammar. (pharmacogenetic map not pharmacogenetics).

Line 174: Check grammar.

Lines 174-175: The sentence does not make any sense.

Line 179: Be careful with the meaning of receptor and the difference with recipients.

Line 180: Be careful with the meaning of receptor and the difference with recipients.

Line 181: check grammar.

Line 183: Check phrasing.

Lines 187-188: Change syntax. Be careful with the meaning of receptor and the difference with recipients.

Lines 188-190: Change syntax, phrasing, grammar.

Line 191: receipt? Check grammar (verb: alter).

Line 192: Be careful with the meaning of receptor and the difference with recipients

Lines 195-197: Change syntax.

Line 198: delete previously.

Lines 199-201: change syntax.

Lines 202-204: check grammar.

Line 208: change syntax.

Line 209: drug-drug  interactions. Change phrasing (length of the event).

Line 211-212: change syntax.

Line 221: Change ΄΄bibliography΄΄to ΄΄references΄΄

The measurement mL: the L is always a capital.

There is a space after a number and before °C and units such as μL, h, min, days, but NOT before % or ° (angle).

General: 

·     The title needs revision

·     There are no keywords

·     References should be in brackets not in parentheses.

·      There is a serious problem in knowledge of basic genetic issues.

·     Typos, spelling, grammar and phrasing issues.

·     Language improvements.

·     Check Punctuation.

·     Check Grammar.

·     Check syntax.

·     All references:  need to be formatted according to journal's style.

·     Some references need to be updated.

·     Some references are not placed in the text ( ref. n. 11, 12, 13).

·     More references should be provided.

Author Response

·         Lines 2-3: The title needs revision.

Answer: Correction accepted.

Line 2-3.... "Frequency of CYP3A5 genetic polymorphisms and Tacrolimus pharmacokinetics in Pediatric Liver Transplantation "

·         Lines 4-5: A * is missing to show the corresponding author.

Answer: Correction accepted. Specific correspondence author

·         Lines 6-13: The affiliations need revision. Emails of the authors are missing. After each affiliation there is no need of commas or dots

Answer: Correction accepted. Lines 6-13:Affiliations were corrected and emails were added

·         Lines 11-13: The correspondence information is not written in the right way

Answer: Correction accepted. Lines 11-13: Corresponding author information corrected

·         Lines 15-27: Abstract does not follow the instructions of the journal. This must be written without headings. 

Answer: Correction accepted. Headings were removed.

·         Line 28: Keywords (3-10) should be included between the Abstract and the Introduction.

Answer: Correction accepted.  Keywords were included between the Abstract and the Introduction

·         Line 37: Check language-grammar. There should not be comma at the rs Number of a SNP.

Answer: Correction accepted,  Line 42. (rs 776746), 

·         Line 38: Check grammar. The allele is associated not the SNP

Answer: Correction accepted,  Line 44: "This allele is associated "

·         Line 41: Non-expressers. Also, non-expressers are homozygous for *3 (CYP3A5*3/*3). Check phrasing.

Answer: Correction accepted, 

·         Line 46-48 :

Answer: Correction accepted, 

The CYP3A5*1 (homozygotes and heterozygotes ) expressers require much higher daily doses of TAC as well as more time to reach desired serum levels of TAC…

·         Lines 44-45: Check phrasing

Answer: Correction accepted, Lines 51-52: After liver transplant  , simultaneous expression of CYP3A5*1 in both the intestine and the implanted liver may occur…

·         Line 55:  Check Punctuation

Answer: Correction accepted. Line 61: Thus it is essential to generate more information to optimize and customize monitoring strategies to liver transplant in this population.

·         Line 56: Check phrasing.

Answer: Correction accepted. Line 63-64 ..The objective of this study was to estimate the frequency of CYP3A5 genetic polymorphisms and their relationship with pharmacokinetics in Pediatric Liver Transplantation..

·         Line 57: Check Punctuation

Answer: Correction accepted. Line 63-64 ..The objective of this study was to estimate the frequency of CYP3A5 genetic polymorphisms and their relationship with pharmacokinetics in Pediatric Liver Transplantation..

·         Line 58: Section 2- Materials and Methods not Methods and Materials.

Answer: Correction accepted, line 66.

·         Line 59,60 : Check phrasing.

Answer: Correction accepted, Line 67 : “A longitudinal study was conducted in f 77 patients under 18 who after liver transplantation over the period 2009-2012 at the Paediatric Hospital Garrahan J. P”

·         Line 61 + 62 + 64 +65: Check phrasing  and Grammar/ syntax..

Answer: Correction accepted, Line 70-75 : “Were included patients with full or partial liver graft, from either living donor or cadaveric donor. All patient were receiving tacrolimus with or without steroids and with or without mofetil mycophenolate. Were excluded HIV infected patients, who suffered early death before receiving immunosuppressive regimen with TAC in the immediate postsurgical and patients with partial or total loss of medical records”

·         Line 67: Check Grammar.

Answer: Correction accepted, Line 77: “Patient information was collected immediately  after  liver transplantation

·         Line 69: patients should not be in capital.

Answer: Correction accepted, Line 779: “for patients after liver transplantation”

·         Line 74: tacrolimus blood levels.

Answer: Correction accepted, Line 85: “”Afterwards the dose of TAC was adjusted to tacrolimus blood level

·         Line 76: ug DNA not DNA ug. Delete initial. Tacrolimus blood levels.

Answer: Correction accepted, Line 85: “4000 copies/ ug DNA) and good renal function, the initial desired TACblood levels were”

·         Line 77: Delete ΄΄:΄΄. Check Punctuation. Check phrasing.

Answer: Correction accepted, Line 87: TAC blood levels were 8-12 ng/ml during the first month after transplantation and then 5-8 ng/ml until

·         Line 79: Numbers 1 to 9 are written in full, except if part of a measurement (6–8 mL) or in the experimental/materials/methods section.

Answer: Correction accepted, Line 88:….. then 5,6,7,8ng/ml until fulfil a year from transplantation

·         Line 80: No meaning.

Answer: Correction accepted, line 91.

·         Line 82: Check Grammar.

Answer: Correction accepted, line 92.

·         Line 95-97-99-100: Check Grammar- Check phrasing (amount- haematological. Red blood cells.-White blood cells. RINàDefine abbreviations.

Answer: Correction accepted, line 106-114

“Demographic information (date of birth, gender), anthropometric data (weight, height), indication of transplant, post- transplantation follow-up time; current medication and doses, concomitant medications, amount of transplanted graft; , amount of postsurgical days, data related to donor type were collected.  We registered clinical laboratory results including hematology (hemoglobin , hematocrit, red blod cells, white cells and platelets, RIN) and clinical chemistry results (creatinine, urea nitrogen, total and direct bilirubin, alkaline phosphatase, alanine aminotransferase- GPT or ALT- aspartate aminotransferase- GOT or AST- gamma glutamyl transpeptidase-GGT- and albumin)”

·         Line 104: Check Grammar.

Answer: Correction accepted, line 116: The genotyping of CYP3A5 were performed in liver biopsies of both the donor and the recipient.

·         Lines 11-112: Check Grammar, phrasing of the whole sentence.

Answer: Correction accepted,  Line  124: Patients with variants (CYP3A5*1/*1 or CYP3A5*1/*3) were called ‘expressers’ while those with variants CYP3A5*3/*3 were called 'not expressers'

·         Line 123: Check Grammar.

Answer: Correction accepted,  Line  136: “Analysis were performed using  STATA 11.0 ©”

·         Line 127: Check grammar.

Answer: Correction accepted, 

·         Line 129:  Check phrasing.

Answer: Correction accepted,  Line 141 : “had at least one acute cellular rejection episode and 8 patients died ( 10.39% ) during follow-up”

·         Line 130-131 : Change syntax- Check phrasing.

Answer: Correction accepted,  Line 141 : “CYP3A5 *1 expression was 37.1% in recipients and 32.2%for Donors”

·         Line 134-135 Check Punctuation. Change receptor expressers to expressers recipients. Check phrasing, Grammar and syntax.

Answer: Correction accepted,  Line 147 : “A total of 3670 blood concentrations of TAC were analysed during the study period, with a mean of 47.8 samples per patient. We observed a greater difference in expressers recipients regarding not expressers, especially..”

·         Line 139: Delete ΄΄-΄΄

Answer: Correction accepted,  Line 152: “A statistically significant reduction in the Co / dose of 0.00063 ug/ml mg/kg/day was observed in comparison with those receiving an organ not expresser..”

·         Line 141: Check Punctuation. A full stop should be placed at the end of the title.

Answer: Correction accepted,  Line 155: “Table 1 Characteristics of the studied population (n=77).

·         Table 1: Check ΄΄origen΄΄, change phrasing (full, technical variant).

Answer: Correction accepted,  Line 155: “ origin”

·         Figure 1: Check grammar. Non-expressers.

Answer: Correction accepted, 

·         Line 145: Check grammar.

Answer: Correction accepted, 

·         Figure 2: Check grammar. Non-expressers.

Answer: Correction accepted, 

·         Line 150: Check grammar, phrasing.

Answer: Correction accepted, Line 155 “The CYP3A5 polymorphisms has a differential impact in the pharmacokinetics of tacrolimus according it expression in donors and recipients”

·         Line 153: Check grammar.

Answer: Correction accepted, Line 158 “In contrast to previous studies this is the first study in the paediatric patients that evaluates the effect of polymorphisms on TAC pharmacokinetics at long term”

·         Line 154: CYP3A5 is a gene, not a variant (do not use rs number).

Answer: Correction accepted, Line 160 “Patients with CYP3A5 allele A (CYP3A5 * 1 or wild type) have a normal splicing of the whole 13 exons in this gene. This results in a normal transcript and producing high levels of mRNA, thus expressing the enzyme metabolizing TAC”

·         Line 155: CYP3A5 is a gene, not a variant (do not use rs number), write genotype.

Answer: Correction accepted

·         Line 158: Delete CYP3A5.

Answer: Correction accepted, Line 162 Patients with allele G (CYP3A5 * 3) have a point mutation (A / G) resulting in the insertion of an inappropriate 'exon' 3B within the transcript.

·         Line 163: references 15, 16, 17 should be placed at the end of the sentence.

Answer: Correction accepted, Line 168: These estimates are consistent with previous results in studies in Argentine renal transplant patients ,which reported values ranging from 9% to 27%(15) (16) (17).

·         Line 165: Check grammar. Reference 18 should be placed at the end of the sentence.

Answer: Correction accepted, Line 171: These differences between the caucasian and asian frequencies, reveal the genetic diversity present in latin america as a result from the colonial stage, African (slaves to the 19th century), and post-independence immigrants (the majority of Spain, Italy, France, Europe from the east) (18).

·         Line 166: East (with capital E).

Answer: Correction accepted, Line 171: These differences between the caucasian and asian frequencies, reveal the genetic diversity present in latin america as a result from the colonial stage, African (slaves to the 19th century), and post-independence immigrants (the majority of Spain, Italy, France, Europe from the east) (18).

·         Line 167: Check grammar. (pharmacogenetic studies not pharmacogenetics).

Answer: Correction accepted, Line 173 “This diversity explained above is also manifested by pharmacogenetic studies

·         Line 169: CYP1A12C, CYP1A21F,CYP3A41B, CYP2D62 are not genes but variants.

Answer: Correction accepted, Line 172:  This diversity explained above is also manifested by pharmacogenetic studies of variants  involved in the metabolism of various antineoplastic agents such as…

·         Lines 170-173: The sentence does not make any sense. Line 173: Check grammar. (pharmacogenetic map not pharmacogenetics).

Answer: Correction accepted, Line 173:  Similar results have been found in studies focused on the frequency of variations in other genes related to antineoplastic metabolism

·         Line 174: Check grammar. Lines 174-175: The sentence does not make any sense.

Answer: Correction accepted, Line 173

Similar results have been found in studies focused on the frequency of variations in other genes related to antineoplastic metabolism (19).  Continue to building this pharmacogenetic map in latin america improve the understanding of the variations in the metabolism and the effect of the different medicines, without the need to extrapolate results obtained from other populations.

·         Line 179: Be careful with the meaning of receptor and the difference with recipients. Line 180: Be careful with the meaning of receptor and the difference with recipients.

Answer: Correction accepted, Line 178

“In liver transplanted patients, both donor and recipient carrying the CYP3A5 polymorphisms are associated with changes in the pharmacokinetics of TAC. However, the role of each polymorphism is different according to days after transplantation. We have shown that the recipient  CYP3A5 genotype plays a more important role than the donor genotype in the early stages either in adult or paediatric population”

·         Line 181: check grammar.

Answer: Correction accepted, Line 181

We have shown that the recipient  CYP3A5 genotype plays a more important role than the donor genotype

·         Line 183: Check phrasing.

Answer: Correction accepted, Line 180

Recipients with  CYP3A5 * 1 achieved lower blood concentrations of TAC and lower dose-adjusted concentrations despite the medical pharmacotherapeutic follow-up (based on adjusting the blood concentrations to the reference therapeutic margins

·         Lines 187-188: Change syntax. Be careful with the meaning of receptor and the difference with recipients.

Answer: Correction accepted, Line 181-2

Recipients with  CYP3A5 * 1 achieved lower blood concentrations of TAC and lower dose-adjusted concentrations despite the medical pharmacotherapeutic follow-up (based on adjusting the blood concentrations to the reference therapeutic margins). These findings are consistent with a recent study of 64 post-transplant children with 1 year follow-up (20).

·         Lines 188-190: Change syntax, phrasing, grammar. Line 191: receipt? Check grammar (verb: alter). Line 192: Be careful with the meaning of receptor and the difference with recipients

Answer: Correction accepted, Line 186

“To recognize the role played by the recipient CYP3A5 genotype in the first weeks after transplantation is essential to avoid excessive dose increases to patients who are expressers of this genotype  (21)”

·         Lines 195-197: Change syntax, Line 198: delete previously.

Answer: Correction accepted, Line 192

This tendency was evidenced in our study; a reduction of dose-adjusted concentrations was documented statistically significant after 60 to 90 postoperative days.

·         Lines 199-201: change syntax.

Answer: Correction accepted, Line 195

Our results indicate the importance of know the genotype present in the organ previously to be implanted. This is a priority during the ambulatory follow-up to select patients with greater hepatic clearance, who will get a lower concentration and which may require different medical follow-up to avoid sub-immunosuppression

·         Lines 202-204: check grammar.

Answer: Correction accepted, Line 201

There biases could be minimized obtaining always information from primary registers (physical or electronic medical history) and checking with other clinical records (nursing records, pharmacy Hospital)

·         Line 208: change syntax, Line 209: drug-drug  interactions. Change phrasing (length of the event). Line 211-212: change syntax.

Answer: Correction accepted, Line 206

We only analyse concentration at time 0 (C0) per patient due to we use data hospital therapeutic monitoring, and for TAC this concentration is used to clinical monitoring for this drugs. Also effect of other variables in the pharmacokinetics of TAC such as  age, drugs interaction and length of the event related to dose-adjusted concentrations were not  evaluated , and should be analyse by other studies

·         Line 221: Change ΄΄bibliography΄΄to ΄΄references΄΄.

Answer: Correction accepted, Line 219

Reviewer 3 Report

The authors presented the results of their study on the impact of CYP3A5 genetic polymorphism on tacrolimus treatment. The main conclusion of the study is that the recipient's status is most important during the first two months after liver transplantation, while the donor's status significantly influences tacrolimus concentrations 90 days after transplantation.

The conclusions of the study are interesting and indicate that both donor's and recipient's CYP3A5 polymorphism could be evaluated before liver transplantation for an optimal dose adjustment.

Unfortunately, the editing of the paper must be improved. There are many errors in the manuscript, several sentences were written oddly, and it is difficult to understand the meaning of some paragraphs. Also, some aspects of methodology and results presentation need elaborating. Below please find a list of major concerns:

In pharmacokinetics, a concentration measured just before a next dose is called a trough concentration or minimal concentration. C0 is a concentration measured in t=0, before the first dose of the drug.

The authors state that in kidney failure or during EBV infection the therapeutic range for tacrolimus concentration was lowered. Please provide data on the number of patients with such an altered regimen. This number must be substantial because the average trough concentration is below the desired 5-8 ng/ml range (Figure 1 and 2). How many patients achieved the targeted TAC concentrations?

Section 2.1 needs to be altered. The fragment about dose adjusting during EBV infection and renal toxicity is vague. Also, what does the "FK" stand for?

Please, provide more background on CY3A5*1 frequency in the African population.

The name of the hospital is spelt differently in every paragraph.

In the Results section, the concentration is provided in ug/ml, whereas in the rest of the manuscript it is expressed in ng/ml. Please, unify this. Also, a reduction cannot be a negative value! The term "reduction" clearly indicates that something is already lower than the value it is compared with.

Figure 1 has an error on X-axis (10 days instead of 30 days). Also, some points do not have error bars.

Author Response

In pharmacokinetics, a concentration measured just before a next dose is called a trough concentration or minimal concentration. C0 is a concentration measured in t=0, before the first dose of the drug

Answer: Correction accepted, line 92

"The values recorded correlate to monitoring blood levels from samples drawn prior to the morning dose (Co) (C0:is a concentration measured in t=0, before the first dose of the drug)".

The authors state that in kidney failure or during EBV infection the therapeutic range for tacrolimus concentration was lowered. Please provide data on the number of patients with such an altered regimen. This number must be substantial because the average trough concentration is below the desired 5-8 ng/ml range (Figure 1 and 2). How many patients achieved the targeted TAC concentrations?

Answer:  

Patients with EBV infection (n=28) have lower TAC concentration , without that difference being statistically significant

Concentration in Patients with EBV infection : 6.85 ng/ml

Concentration in Patients without EBV infection : 6.69 ng/ml , p=0.098

Also , only 3 patients had GFR less than 60 (ml/min/1.73m2) without that differences in TAC concentrations statistically significant

With respect to the number of patients who reached the target levels, this is relative given that this was a longitudinal and not a cross-sectional study. Although a post-surgical patient did not reach levels during their follow-up, they could reach it and perhaps at some point during the two years their levels went out of range. Precisely this is the wealth of this work that shows a behavior over time and not a classification in a single moment of time.

Section 2.1 needs to be altered. The fragment about dose adjusting during EBV infection and renal toxicity is vague. Also, what does the "FK" stand for?

Answer:  Correction accepted, line 84

"The initial oral regimen was 0.1 mg/kg/day every 12 hours. Afterwards the dose of TAC was adjusted to tacrolimus blood levels, liver parameters, kidney function and the viral load of Epstein Barr Virus (10). In patients without infectious activity (viral load less than 4000 copies/ ug DNA) and creatinine clearance less than the expected range for your age, the initial desired TACblood levels were 8-12 ng/ml during the first month after transplantation and then 5,6,7,8ng/ml until fulfil a year from transplantation (10). It was proceeded a quick immunosuppression reduction in patients with viral load above 4000 copies / ug DNA in 2 consecutive samples or clinical evidence of EBV infection.. No antiviral therapy was implemented. In patients who developed renal toxicity, regardless viral load, monitoring of TAC  was decreased a 25%. In those cases mycophenolate mofetil (MMF) was added as rescue therapy with an initial dose of 20 mg/kg/day and then it was increased up to 40 mg/kg/day after a week of treatment"

Please, provide more background on CY3A5*1 frequency in the African population

Answer:  There must be an error in the reviewer. The study was done in Latin American, not African, population. There are no previous studies to our respect for this variane in Latin American population

The name of the hospital is spelt differently in every paragraph.

Answer:  Correction accepted in all text.

In the Results section, the concentration is provided in ug/ml, whereas in the rest of the manuscript it is expressed in ng/ml. Please, unify this. Also, a reduction cannot be a negative value! The term "reduction" clearly indicates that something is already lower than the value it is compared with

Answer:  Correction accepted, line 154 ug/m replaced by ng/ml

Figure 1 has an error on X-axis (10 days instead of 30 days). Also, some points do not have error bars.

Answer:  Correction accepted

Round 2

Reviewer 1 Report

The topic is very interesting, studying the effect of donor liver on the PK of tacrolimus appears to be a relevant topic.

However the article is really difficult to read, not only because of mistakes in language.

The paper really needs major re-writing with accurate description of methods and of results.

Author Response

Correction accepted

Reviewer 2 Report

In this revised version the authors have adequately addressed my comments

Author Response

correction accepted

Reviewer 3 Report

Please, provide more background on CY3A5*1 frequency in the African population

Answer:  There must be an error in the reviewer. The study was done in Latin American, not African, population. There are no previous studies to our respect for this variane in Latin American population

Reply: I am aware that the paper concerns the Latin American population. In lines 45-46, the authors list frequencies in Caucasian and Asian populations. Therefore I think that the abstract should also include frequencies in other populations, for the sake of completeness.

Author Response

 I am aware that the paper concerns the Latin American population. In lines 45-46, the authors list frequencies in Caucasian and Asian populations. Therefore I think that the abstract should also include frequencies in other populations, for the sake of completeness.

-It is correct, the comparative discussion of the frequencies reported in the previous studies and ours is in line 192, given the limitation in words of the abstract. 

Round 3

Reviewer 3 Report

The paper has been significantly improved and now meets the criteria for publication.